# Effectiveness of ERAS (Enhanced Recovery after Surgery) Protocol via Peripheral Nerve Block for Total Knee Arthroplasty

**DOI:** 10.3390/jcm11123354

**Published:** 2022-06-10

**Authors:** Hyun Hee Lee, Hyuck Min Kwon, Woo-Suk Lee, Ick Hwan Yang, Yong Seon Choi, Kwan Kyu Park

**Affiliations:** 1Department of Orthopedic Surgery, International St. Mary’s Hospital, Catholic Kwandong University College of Medicine, Incheon 22711, Korea; emprany@ish.ac.kr; 2Department of Orthopedic Surgery, Yonsei University College of Medicine, Seoul 03722, Korea; 3Department of Orthopedic Surgery, Severance Hospital, Yonsei University College of Medicine, Seoul 03722, Korea; hyuck7777@yuhs.ac; 4Department of Orthopedic Surgery, Gangnam Severance Hospital, Yonsei University College of Medicine, Seoul 06273, Korea; wsleeos@yuhs.ac; 5Department of Orthopedic Surgery, Seran Hospital, Seoul 03030, Korea; ihyang@yuhs.ac; 6Department of Anesthesiology and Pain Medicine, Severance Hospital, Yonsei University College of Medicine, Seoul 03722, Korea

**Keywords:** peripheral nerve block, femoral nerve block, adductor canal block, patient-controlled analgesia, total knee arthroplasty, enhanced recovery after surgery

## Abstract

Peripheral nerve block (PNB) for patients with total knee arthroplasty (TKA) is one of the recommended interventions in ERAS protocols. However, most existing studies involved unilateral TKA (UTKA). As such, this study aimed to evaluate the effectiveness of PNB in terms of immediate postoperative analgesia, length of hospital stays (LOS), and early functional outcomes in both UTKA and simultaneous bilateral TKAs (BTKAs). We reviewed 236 patients who underwent primary TKA with PNB, with 138 and 98 being UTKA and BTKAs, respectively; those in the PNB group underwent femoral nerve and adductor canal block. The matched control and PNB groups—who received intravenous/epidural patient-controlled analgesia (IVPCA/PCEA) alone or IVPCA in addition to PNB after surgery, respectively—were compared. The VAS scores at rest until 48 h after surgery were significantly lower in PNB groups compared to those in the IVPCA groups. At 0– 6 h of activity, VAS scores of the UTKA with PNB group were also lower than the IVPCA group. Compared to PCEA groups, VAS scores at 0–6 h of activity were higher in both the UTKA and BTKAs with PNB groups. However, at 24–48 h at rest, the scores of those in the UTKA with PNB group were lower than those in the PCEA group. The control and experimental UTKA and BTKAs groups had similar LOS and functional outcomes at 90 days postoperatively. In primary TKA, PNB has great analgesic effects for immediate postoperative pain control, and represents a similar analgesic effect to epidural PCA.

## 1. Introduction

Osteoarthritis (OA) is the most common rheumatic disease of the musculoskeletal system, with the knee being the most commonly affected joint. The predominant clinical symptoms of knee OA are pain, joint stiffening, crepitus [1], inflammation, and muscle weakness [2]. Established conservative treatments for osteoarthritis include exercise [3], knee bracing [4], physical modalities [5], and pharmacology [6], but their long-term effectiveness is limited, thus total knee replacement is ultimately needed for the majority of patients with knee OA.

Total knee arthroplasty (TKA) is one of the most common and successful orthopedic procedures to treat patients with severe knee osteoarthritis. However, patients who undergo TKA often experience severe pain after surgery. Therefore, postoperative pain control in TKA has been the most crucial challenge for orthopedic surgeons [7]. Severe pain produces a prolonged length of hospital stay (LOS), low patient satisfaction, and increased opioid consumption, which can potentially elicit side effects such as gastrointestinal problems, altered cognitive function, urinary retention, pruritus, and respiratory depression [8,9,10,11]. To manage these consequences, enhanced recovery after surgery (ERAS), first described by Henrik Kehlet in 1997 [12], has begun to be discussed in the surgical field. The concepts of ERAS are being continuously investigated in the field of orthopedics.

Peripheral nerve block (PNB) for patients with TKA is a recommended intervention in ERAS protocols. Femoral nerve block (FNB), the most common PNB method, is reportedly effective in decreasing pain and facilitating early rehabilitation after TKA [13,14]; further, it provides sufficient analgesia with fewer side effects, including neurological complications, vomiting, and nausea, compared to PCEA or IVPCA with opioids [7,14,15]. However, a disadvantage of FNB is the impairment of motor function, thus delaying the restoration of quadriceps strength.

Adductor canal block (ACB), an alternative nerve block technique, selectively blocks the sensory branch of the femoral nerve. Given that the adductor canal is placed in the middle third of the thigh, runs from the apex of the femoral triangle proximally to the adductor hiatus distally, and consistently encloses the saphenous nerve and the nerve to the vastus medialis, ACB can spare the major motor branches of the femoral nerve [16,17]. Resultantly, this can relieve pain without weakening the quadriceps. Based on recent studies, PNBs, such as FNB and ACB, may significantly reduce pain after TKA. However, most existing studies involved UTKA, and the comparison between PNB versus epidural or intravenous PCA after both UTKA and BTKAs appears insufficient. It is important as part of the ERAS program to find out how effective PNB is for both UTKA and BTKAs.

Therefore, this study asked the following questions. Compared to PNB and PCA only, (1) how effective was postoperative pain control after both UTKA and BTKAs, and (2) is there any difference in LOS and functional outcomes in the early postoperative period? Finally, (3) which procedure is more effective compared to FNB and ACB? By finding the answers to these questions, it will be possible to apply appropriate postoperative management to patients who undergo TKA. Moreover, based on these studies, we will provide protocols that can be newly applied to the ERAS program.

## 2. Materials and Methods

### 2.1. Study Design

This retrospective comparative study with a propensity score matching analysis was approved by the Institutional Review Board (4-2021-0772) of the authors’ facility. The medical records of patients who underwent primary TKA for knee osteoarthritis after performing an imaging examination at a single center were retrospectively reviewed.

### 2.2. Participants

A total of 857 consecutive TKAs were performed by a single surgeon between March 2015 and March 2021. Only patients for whom all data elements were prospectively collected—including demographics, anesthesia type, and American Society of Anesthesiologists (ASA) class—were included. Patients with ASA class ≥ 4; with secondary arthritis due to rheumatoid arthritis or trauma; who needed special instrumentation due to severe instability, bone defect, or anatomical deformity; who required additional procedures other than TKA; or had a history of revision surgery were excluded. In this study, 1:1 propensity score matching was used to minimize selection bias, with matched variables being age at operation, gender, BMI, and ASA class. After matching, 138 cases with PNB were matched with 138 cases with PCA alone in the unilateral TKA group, whereas 98 cases with PNB were matched with 98 cases with PCA alone in the simultaneous bilateral TKAs group (Figure 1).

### 2.3. Surgery

All operations were performed according to the standard protocol under pneumatic tourniquet inflation, with cemented posterior-stabilized (PS)-type TKA prostheses from nine manufacturers. For median skin incisions, a standard medial parapatellar arthrotomy approach was used, and patella resurfacing was not done. All patients started active and passive knee range of motion (ROM) exercises one day postoperatively under the same rehabilitation protocol.

### 2.4. Definition of Groups

The PNB group was defined as cases undergoing TKA under general anesthesia or spinal anesthesia with PNBs. The PNBs included femoral nerve block and adductor canal block. In UTKA, PNB was performed with continuous FNB (CFNB) or continuous ACB (CACB). In BTKAs, the nerve block was performed on each knee by a combination of CFNB + single-shot FNB (SSFNB), CFNB + single-shot ACB (SSACB), CACB + SSFNB, CACB + SSACB, and CACB + CACB. After the TKA, all PNBs were performed by an anesthetist. In FNBs, a femoral catheter was inserted in the femoral canal just below the inguinal ligament. ACB was performed at the mid-thigh level using an ultrasound transducer. The regimen of continuous PNB was an infusion of 0.2% ropivacaine 6 mL per hour, for a total volume of 280 mL. For single-shot PNB, 20 mL of 0.2% ropivacaine was injected. After the PNB was performed, intravenous PCA was added in all PNB group patients.

The PCA group consisted of intravenous PCA (IVPCA) and patient-controlled epidural analgesia (PCEA). Fentanyl 10 μg/kg was used in IVPCA. A PCEA device was connected to an epidural catheter, and 0.15% ropivacaine was infused when sensory levels dropped below T12. In the PCA group, the patient received either IVPCA or PCEA alone.

All patients routinely received 200 mg of celecoxib by mouth preoperatively. Postoperatively, patients received 200 mg of celecoxib daily. For breakthrough pain, “rescue medication” included tramadol 50–100 mg IV and acetaminophen 10 mg IV every four hours as needed.

### 2.5. Outcome Measures

The primary outcome was pain intensity score, which was measured on a visual analog scale (VAS; rated from 0–10, where 0 = no pain and 10 = worst possible pain). VAS scores at rest and activity were monitored during the first 48 h after surgery at 3 intervals: 0 to 6 h, 6 to 24 h, and 24 to 48 h. Secondary outcomes—hospital LOS (days) and, for the determination of early functional outcomes, American Knee Society (AKS) scores and Western Ontario and McMaster Universities Osteoarthritis (WOMAC) index—were evaluated preoperatively and at the 90-day postoperative follow-up.

### 2.6. Propensity Score Matching

We used propensity score-matched analyses to reduce the selection bias and potential baseline differences between the PNB and PCA groups. Propensity scores were calculated using a logistic regression model in which the dependent variable was whether the patient was given a PNB. The independent variables were age, sex, BMI, and ASA class. These variables were selected according to previous reports [18,19].

### 2.7. Statistical Analysis

Descriptive statistics were performed, and normality distribution analysis was assessed by the Shapiro–Wilk test. Continuous variables were analyzed using the Student’s *t*-test or Mann–Whitney test for normal and non-normal distributions, respectively. Categorical variables were compared using the chi-squared test. For the evaluation of non-normal distributions of the continuous variables of multiple groups, the Kruskal–Wallis test was performed. The Bonferroni correction method was also performed for post-hoc analysis. Data analysis was conducted using the Statistical Package for the Social Sciences software, version 25.0 (SPSS, IBM Inc., Chicago, IL, USA).

## 3. Results

The baseline characteristics of patients in the PNB and PCA groups were comparable in terms of age, gender, BMI, ASA class, and types of anesthesia (Table 1).

### 3.1. Primary Outcome

The VAS score was compared as the primary outcome. As a result, the VAS scores 0–48 h postoperatively during rest were lower in the PNB group than in the IVPCA group for both UTKA and BTKAs. At 0–6 h postoperatively, during activity, the VAS scores of the PNB group with UTKA were also lower. The VAS scores among those who underwent a UTKA were lower in the PNB group than in the IVPCA group at rest 0–6, 6–24, and 24–48 h postoperatively (*p* < 0.001, 0.003, and 0.001, respectively), and 0–6 h postoperatively during activity of the knee (*p* = 0.008). In BTKAs, the VAS scores of the PNB group at rest 0–6, 6–24, and 24–48 h postoperatively (*p* < 0.001, 0.008, and 0.001, respectively) were lower than in the IVPCA group (Figure 2 and Figure 3).

Compared to the PCA and PCEA groups, among those who underwent a UTKA, the VAS scores of the PNB group at rest 24–48 h postoperatively were lower (*p* < 0.001). However, the pain scores of the PCEA group with both UTKA and BTKAs at 0–6 h postoperatively during activity were lower (*p* = 0.043 and 0.039, respectively) than those in the PNB group (Figure 4 and Figure 5). There was no statistically significant difference in the VAS scores among groups with UTKA and BTKAs at the other monitored periods.

### 3.2. Secondary Outcomes

In general, the PNB and PCA groups did not differ in terms of hospital LOS or early functional outcomes, represented by AKS scores and WOMAC index (Table 2). However, preoperatively, the AKS function scores of the UTKA and BTKAs groups differed significantly (*p* = 0.009 and <0.001, respectively).

### 3.3. Subgroup Analysis

Subgroup analysis was also conducted to estimate the analgesic effect among PNB procedures. In patients with UTKA, there were no statistically significant differences in the VAS scores of the CFNB and CACB groups 48 h postoperatively (Table 3). Compared to patients who underwent BTKAs, the VAS scores of the CFNB + SSFNB group at 6–24 h and 24–48 h postoperatively during rest were lower than those of the CACB + SSFNB group (*p* = 0.002 and 0.001, respectively). The difference in pain intensity for each postoperative period compared to the other groups was similar (Table 4).

## 4. Discussion

The present study is a large-sample trial that evaluated the analgesic effect, hospital LOS, and early functional outcomes of PNB compared with PCA in patients undergoing UTKA and BTKAs. It found that there are significant differences in pain scores during the first 48 h after surgery. The VAS scores 0–48 h postoperatively during rest were lower in the PNB group than in the IVPCA group for both UTKA and BTKAs. At 0–6 h postoperatively, during activity, the VAS scores of the PNB group with UTKA were also lower. Compared to the PCEA group, pain scores were higher during activity 0–6 h postoperatively, although scores at rest 24–48 h postoperatively among those who underwent UTKA were lower. From these results, it was found that a very satisfactory analgesic effect was obtained when PNB was added to IVPCA, and comparable results were obtained compared with PCEA.

Severe pain after TKA has been shown to affect functional recovery [20]. Therefore, post-TKA pain control is an essential consideration. Various studies have evaluated the effectiveness of PNBs for postoperative analgesia compared with IVPCA [14,21]. In this context, a meta-analysis by Paul et al. [22] reported that single-shot and CFNB was superior to PCA alone. In particular, CFNB with IVPCA has better outcomes compared to PCA alone regarding reduced morphine consumption, pain scores, and nausea [23]. In this study, PNB—consisting of FNB and ACB—was compared with IVPCA alone; similar results in postoperative pain control were obtained. Although not documented, it is expected that opioid consumption can also be reduced by adding PNB to IVPCA.

PNB provides intense site-specific analgesia and has lesser side effects compared to epidural techniques [24]. Fowler et al. [25] reported PNB to have a reduced side-effect profile, such as less urinary retention and hypotension, than that of epidural analgesia, while representing a similar analgesic effect. Patients treated with a unilateral nerve block also experienced less restriction of motor function than those who received epidural analgesia [8]. Similarly, Barrington et al. [26] observed equivalent analgesia between CFNB and CEA groups after TKA; however, the regimen of each group was different (0.2% bupivacaine for femoral infusion, and 0.2% ropivacaine with 4 mcg/mL fentanyl for epidural infusion). In this study, 0–6 h postoperatively, both during rest or activity, VAS scores were significantly higher (or comparable) in the PNB group than the PCEA group, mainly due to the inability of the femoral approach to block sciatic and obturator components [24]. After this period, the pain scores became similar; in particular, the VAS score of the PNB group with UTKA was lower during activity 6–24 h postoperatively.

Hospital LOS and early functional outcomes were also evaluated as secondary outcomes in this study. Hospital LOS reflects the economic burden of each patient [27]. Furthermore, hospital LOS is dependent on many factors, including preoperative hemoglobin, age, and gender [28,29]. The current study shows that there is no statistically significant difference between PNB and PCA groups in terms of hospital LOS. Several studies similarly reported no difference in hospital LOS [30,31]. At present, there is no conclusive evidence that PNB could reduce hospital LOS compared to PCA alone. To the researchers’ knowledge, there is a lack of research comparing early follow-up results in functional outcomes between PNB and PCA alone. In this study, medical records measuring AKS score and WOMAC index (preoperatively and 90 days postoperatively) were collected and compared to assess early functional outcomes; this study found that there were no significant differences in functional outcomes, except for preoperative differences in AKS function scores, in both UTKA and BTKAs. The AKS score is divided into two categories, knee and function, and each category consists of detailed subjects measuring pain, range of motion, stability, quadriceps muscle power, and so on [32]. The WOMAC index also consists of subjects that measure pain, function, and stiffness for various situations [33]. There have been several studies of patient function in the immediate postoperative period. Chan et al. [34] found that patients receiving FNB had increased range of motion (MD = 6.48 degrees, 95% CI = 4.27 to 8.69 degrees) and higher patient satisfaction (SMD = 1.06, 95% CI = 0.74 to 1.38) compared with patients who received IVPCA with no PNB. Beaupre et al. [35] found similar mobility between patients who received PCA alone and those who received PCA and PNB. In the present study, we suggested that there was no difference in functional outcomes at 90 days after surgery through a comparison of AKS score and WOMAC index.

A meta-analysis by Gao et al. [36] reported that ACB showed faster ambulation ability recovery after TKA compared with FNB, with no significant difference in postoperative pain control and opioid consumption in the early postoperative period. In this study, although quadriceps strength was not quantitatively measured in all cases, there were no reports of significant motor weakness in PNB groups. Regarding pain control, several studies have also reported no significant difference between ACB and FNB groups [16,37,38,39,40]. In the present study’s subgroup analysis, a comparison of CFNB and CACB patients with UTKA showed no significant difference in analgesic effect. Patients with BTKAs received five combinations of blocks for each knee according to the method of PNB, which also showed similar pain control effects. The similar analgesic effect between the FNB and ACB groups may be explained by how most nerves in the adductor canal are sensory nerves dominating knee joints. Therefore, ACB seems to not compromise pain relief, and may allow more preservation of quadriceps muscle strength than FNB with comparable pain control [36,37]. The nerve block method applied to each knee in patients who underwent BTKAs was divided into five combinations to consider cost-effectiveness. The insurance policy issue of the country (Korea, Republic of) to which our institution belongs was also a consideration.

In this study, we found that a very satisfactory analgesic effect was obtained when PNB was added to IVPCA, and comparable results were obtained compared with PCEA. Based on these results, it is thought that PNB can be actively applied clinically not only in UTKA, but also in BTKA. Additionally, PNB was shown to have a better effect on pain control in UTKA than BTKAs. The cause of these results can be thought to be that, in the case of BTKAs, the continuous dose and the single shot block were applied to each knee, respectively. According to previously reported case series [41,42], in BTKAs, when continuous nerve blocks were performed on both sides, it was effective in postoperative pain control. Considering the analgesic effect of PNB, it is thought that it can be applied to revisional TKA, which will contribute to the expansion of the scope of application of PNB. Therefore, further is needed in this regard.

The present study has several limitations. First, are the inherent limitations of the retrospective comparative design of this study. There was also difficulty in controlling all factors affecting outcomes. To counteract possible bias, this study analyzed relatively large cases of both UTKA and BTKAs. Further, propensity score matching was performed to match the baseline characteristics of patients [43]. Second, was the nonuniformity of the TKA implant manufacturer. There were nine brands, but they all had the same cemented PS-type prosthesis design. Finally, there was a lack of measurement of postoperative functional aspects, including mobilization, muscle strength, and range of motion.

## 5. Conclusions

In summary, PNB has great analgesic effects for immediate postoperative pain control after total knee arthroplasty. Even compared to PCEA, PNB represents a similar analgesic effect. There is no significant difference in hospital LOS and early functional outcomes between PNB with PCA and PCA alone. Finally, a comparison of FNB and ACB showed similar pain control. We suggest PNB as a component of an ERAS protocol for patients undergoing total knee arthroplasty that could relieve immediate postoperative pain and encourage a return to daily activities. To determine which of the FNB or ACB is effective, further investigations are needed considering the prospective, larger sample size, and cost-effectiveness of these anesthesia modalities.

## Figures and Tables

**Figure 1 jcm-11-03354-f001:**
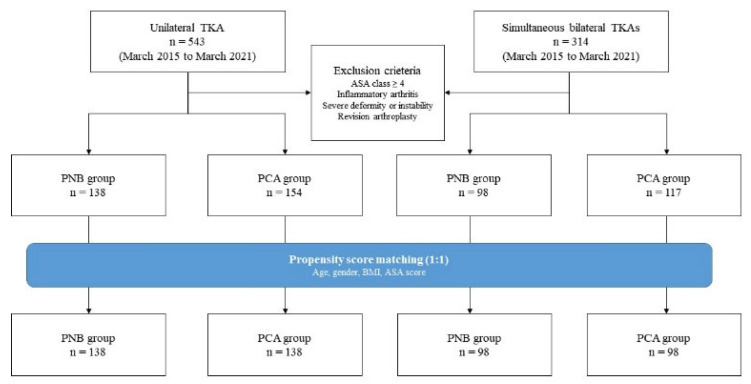
Flowchart of patient selection in the study is shown.

**Figure 2 jcm-11-03354-f002:**
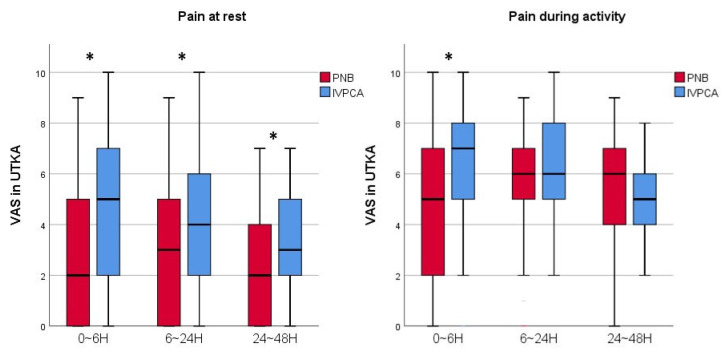
A graph showing the VAS scores of the PNB and IVPCA groups with UTKA. Horizontal lines represent medians; boxes represent 25th to 75th percentiles; whiskers represent minimums and maximums. * Significant difference between two groups was expressed. UTKA = unilateral total knee arthroplasty; PNB = peripheral nerve block; IVPCA = intravenous patient-controlled analgesia; VAS = visual analogue scale; H = hours.

**Figure 3 jcm-11-03354-f003:**
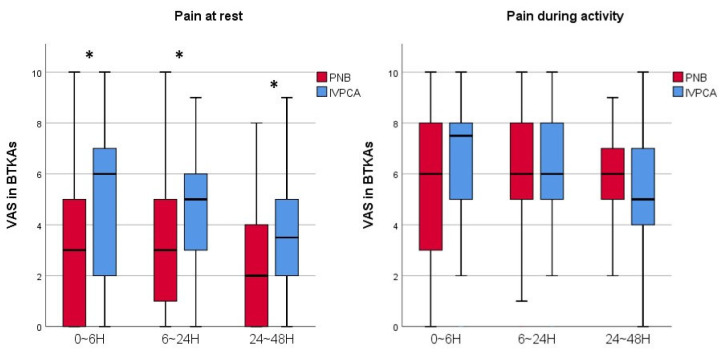
A graph showing the VAS scores of the PNB and IVPCA groups with BTKAs. Horizontal lines represent medians; boxes represent 25th to 75th percentiles; whiskers represent minimums and maximums. * Significant difference between two groups was expressed. BTKAs = simultaneous bilateral total knee arthroplasty; PNB = peripheral nerve block; IVPCA = intravenous patient-controlled analgesia; VAS = visual analogue scale; H = hours.

**Figure 4 jcm-11-03354-f004:**
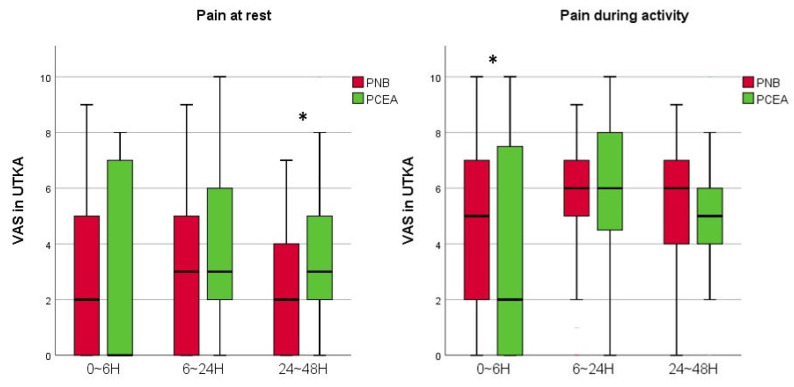
A graph showing the VAS scores of the PNB and PCEA groups with UTKA. Horizontal lines represent medians; boxes represent 25th to 75th percentiles; whiskers represent minimums and maximums. * Significant difference between two groups was expressed. UTKA = unilateral total knee arthroplasty; PNB = peripheral nerve block; PCEA = patient-controlled epidural analgesia; VAS = visual analogue scale; H = hours.

**Figure 5 jcm-11-03354-f005:**
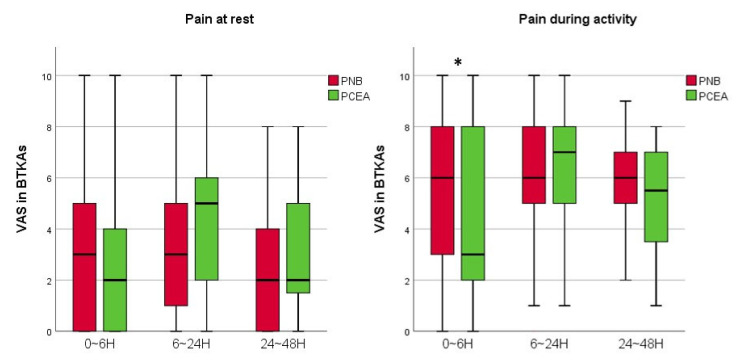
A graph showing the VAS scores of the PNB and PCEA groups with BTKAs. Horizontal lines represent medians; boxes represent 25th to 75th percentiles; whiskers represent minimums and maximums. * Significant difference between two groups was expressed. BTKAs = simultaneous bilateral total knee arthroplasty; PNB = peripheral nerve block; PCEA = patient-controlled epidural analgesia; VAS = visual analogue scale; H = hour.

**Table 1 jcm-11-03354-t001:** Baseline characteristics of the patients undergoing TKA.

	UTKA				BTKAs		
	PNB Group (n = 138)	PCA Group (n = 138)	*p*-Value		PNB Group (n = 98)	PCA Group (n = 98)	*p*-Value
Age	71.3 ± 6.6	70.3 ± 6.9	0.244		71.2 ± 6.0	70.6 ± 5.5	0.487
Gender, No. (%)			0.084				0.578
Male	30 (21.7)	19 (13.8)			16 (16.3)	19 (19.4)	
Female	108 (78.3)	119 (86.2)			82 (83.7)	79 (80.6)	
BMI	26.4 ± 3.2	26.3 ± 3.4	0.821		27.0 ± 3.9	26.7 ± 3.7	0.566
ASA			0.918				0.642
1	6	7			9	5	
2	66	63			47	51	
3	66	68			42	42	
PNB method							
CFNB	49	NA		CFNB + SSFNB	42	NA	
CACB	89	NA		CFNB + SSACB	8	NA	
				CACB + SSFNB	21	NA	
				CACB + SSACB	22	NA	
				CACB + CACB	5	NA	
PCA							
IV	NA	82			NA	54	
Epidural	NA	56			NA	44	
PNB + IV	138	NA			98	NA	
Anesthesia			0.326				0.456
General	50	58			53	44	
Spinal	88	80			45	54	

Data are shown as mean ± standard deviation for normally distributed variables. UTKA = unilateral total knee arthroplasty; BTKAs = simultaneous bilateral total knee arthroplasty; BMI = body mass index; ASA = American society of anesthesiologists; PNB = peripheral nerve block; CFNB = continuous femoral nerve block; CACB = continuous adductor canal block; SSFNB = single-shot femoral nerve block; SSACB = single-shot adductor canal block; PCA = patient-controlled analgesia; IV = intravenous; NA = not applicable.

**Table 2 jcm-11-03354-t002:** Comparison of hospital LOS and functional outcomes (measured by AKS scores and WOMAC index) between PNB and PCA groups.

	UTKA				BTKAs		
	PNB Group (n = 138)	PCA Group (n = 138)	*p*-Value		PNB Group (n = 98)	PCA Group (n = 98)	*p*-Value
Hospital LOS	5.3 ± 1.3	5.2 ± 0.8	0.173		5.3 ± 1.4	5.4 ± 1.5	0.630
AKS knee score							
Preoperatively	52.5 ± 15.6	52.9 ± 19.0	0.857		50.9 ± 11.3	50.6 ± 17.2	0.876
Postoperatively 3 M	87.3 ± 17.7	84.0 ± 17.1	0.171		87.7 ± 16.7	89.6 ± 11.9	0.217
AKS function score							
Preoperatively	60.6 ± 16.7	55.3 ± 17.2	0.009 *		60.1 ± 14.6	50.9 ± 20.1	<0.001 *
Postoperatively 3 M	75.5 ± 18.2	73.4 ± 19.9	0.405		76.1 ± 15.7	75.5 ± 19.4	0.756
WOMAC							
Preoperatively	53.3 ± 18.7	52.7 ± 21.1	0.797		55.7 ± 19.5	54.4 ± 25.5	0.550
Postoperatively 3 M	25.5 ± 25.2	26.7 ± 15.2	0.645		25.6 ± 22.6	25.8 ± 18.3	0.897

Data are shown as mean ± standard deviation for normally distributed variables. * Significant difference between two groups. UTKA = unilateral total knee arthroplasty; BTKAs = simultaneous bilateral total knee arthroplasty; LOS = length of stay; M = months; AKS = American Knee Society; WOMAC = Western Ontario and McMaster Universities Osteoarthritis Index.

**Table 3 jcm-11-03354-t003:** Subgroup analysis of VAS scores between CFNB and CACB groups in patients with UTKA.

	CFNB (n = 49)	CACB (n = 89)	*p*-Value
0–6 h Rest	1 (0,4)	2 (0,5)	0.252
0–6 h Activity	4.5 (2,7)	5 (3,7)	0.208
6–24 h Rest	2.5 (0,5)	3 (1,5)	0.152
6–24 h Activity	6 (4,7)	6 (5,7)	0.373
24–48 h Rest	2 (0,4)	2 (0,3)	0.613
24–48 h Activity	5 (3,6)	6 (4,7)	0.138

Data are shown as median and interquartile ranges for variables that were not normally distributed. VAS = visual analogue scale; CFNB = continuous femoral nerve block; CACB = continuous adductor canal block; h = hours.

**Table 4 jcm-11-03354-t004:** Subgroup analysis of VAS scores among PNB groups in patients with BTKAs.

	CFNB + SSFNB (n = 42)	CFNB + SSACB (n = 8)	CACB + SSFNB (n = 21)	CACB + SSACB (n = 22)	CACB + CACB (n = 5)	*p*-Value
0–6 h Rest	3 (0,4)	3.5 (1,5)	2 (0,5)	2.5 (0,5)	6 (2,8)	0.533
0–6 h Activity	6 (3,7)	5 (4,7.5)	7 (3,8)	4.5 (3,8)	8 (6,9)	0.302
6–24 h Rest	2 (0,5)	3 (0,6.5)	5 (3,6)	3 (2,6)	6 (3,8)	0.008 *
6–24 h Activity	6 (5,7)	6 (4,8)	7 (6,8)	6.5 (5,8)	8 (7,9)	0.058
24–48 h Rest	2 (0,3)	1 (0,3.5)	4 (2,5)	2 (0,3)	3 (3,3)	0.010 *
24–48 h Activity	6 (5,7)	4.5 (3,6.5)	6 (5,7)	5 (3,7)	6 (5,6)	0.617

Data are shown as median and interquartile ranges for variables that were not normally distributed. * Significant difference between two groups; VAS = visual analogue scale; CFNB = continuous femoral nerve block; CACB = continuous adductor canal block; SSFNB = single-shot femoral nerve block; SSACB = single-shot adductor canal block; h = hours.

## Data Availability

Data are the property of the authors and are available by contacting the corresponding author.

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
