# Peer review of "Effectiveness of ERAS (Enhanced Recovery after Surgery) Protocol via Peripheral Nerve Block for Total Knee Arthroplasty"

_jcm, 2022, doi:10.3390/jcm11123354_

Round 1

Reviewer 1 Report

The author retrospectively compared the effectiveness of ERAS in TKA surgery between PNB and their matched cohort. In general, this paper is interesting and impactful. I have some concerns.

1.  Matching criteria for propensity score matching analysis were age at operation, gender, BMI, and ASA class. Why did the authors select these 3 criteria and exclude other characteristics? 
2. Following, were there any differences in demographic between PNB and PCA group before matching?
3. Did PNB group receive analgesic agents for pain control in the perioperative period?
4. The analgesic agent consumption post-TKA period should be presented.
5. PNB works better in UTKA than STKA. The author should address this issue in the discussion section.

Author Response

Response to Reviewer 1 Comments

Point 1: Matching criteria for propensity score matching analysis were age at operation, gender, BMI, and ASA class. Why did the authors select these 3 criteria and exclude other characteristics? 

Response 1: First of all, we really thank the reviewer for taking time and making effort to review our paper. To reduce the impact of selection bias on study outcomes, we used propensity score–matched-pair analyses to determine the adjusted association of peripheral nerve block with the primary (VAS) and secondary outcomes. The rationale and methods underlying the use of propensity scores for proposed causal exposure variables in the context of cohort studies have been previously described1. Because age at operation, gender, BMI and ASA class are the most objective and generally matched factors measured in all patients, these factors were set as matching criteria. We have added a relevant explanation for this to the manuscript. Change can be seen in line 137-142.

Reference:

  1. Rubin DB: The design versus the analysis of observational studies for causal effects: Parallels with the design of randomized trials. Stat Med 2007; 26:20–36

Point 2:  Following, were there any differences in demographic between PNB and PCA group before matching?

Response 2: Although there was no significant difference in baseline patient characteristics before matching, it can be said that matching was meaningful in minimizing selection bias.

Point 3:  Did PNB group receive analgesic agents for pain control in the perioperative period?

Response 3: The same analgesic agents were used in both the PNB and PCA groups. All patients routinely received 200 mg of celecoxib by mouth preoperatively. Postoperatively, patients received 200 mg of celecoxib daily. For breakthrough pain, “rescue medication” included tramadol 50-100 mg IV and acetaminophen 10 mg IV every four hours as needed.

Point 4:  The analgesic agent consumption post-TKA period should be presented.

Response 4: As explained above, postoperatively, patients received 200 mg of celecoxib daily. For breakthrough pain, “rescue medication” included tramadol 50-100 mg IV and acetaminophen 10 mg IV every four hours as needed. These regimens remained the same until the patient was discharged. Change can be seen in line 125-128.

Point 5:  PNB works better in UTKA than STKA. The author should address this issue in the discussion section.

Response 5: Thanks for the great suggestion from the reviewer. In consideration of this suggestion, a relevant matter has been added to the discussion section. The cause of these results can be thought to be that in the case of BTKAs, the continuous dose and the single shot block were applied to each knee, respectively. Change can be seen in line 313-323.

Reviewer 2 Report

ABSTRACT

The link between background information and study aims is not clear.

INTRODUCTION

Please start the first sentence in the following way and cite suggested papers:

“Osteoarthritis (OA) is the most common rheumatic disease of the musculoskeletal system, with the knee being the most commonly affected joint . The predominant clinical symptoms of knee OA are pain, joint stiffening, creptitus (https://pubmed.ncbi.nlm.nih.gov/12603937/), inflammation and muscle weaknes (https://pubmed.ncbi.nlm.nih.gov/28929165/ ). Established conservative treatement for osteoarthritis include exercise (https://www.ncbi.nlm.nih.gov/pmc/articles/PMC3635671/ ), knee bracing (https://pubmed.ncbi.nlm.nih.gov/29931372/ ), physical modalities  ( https://pubmed.ncbi.nlm.nih.gov/25162407/ ), pharmacology (https://www.ncbi.nlm.nih.gov/pmc/articles/PMC6315310/ ), but their long-term effectiveness is limited, thus total knee replacement is ultimately needed for the majority of patients with knee OA. “

In the current form, it is quite difficult to figure out from the information flow in the introduction, why it is important to study this, who will benefit from it, and what is the added value of this paper to current knowledge.

METHODS

The whole methods section should be rewritten according to relevant reporting guidelines, such as STROBE or CONSORT. Information on study design, setting, inclusion and exclusion criteria for study participants, the definition of outcomes, validity and reliability of instruments, data postprocessing etc is very superficial and should be provided in separate paragraphs to facilitate reading.

RESULTS

Can authors better structure the results section? Given the amount of information, I would suggest providing subsections based on the outcomes or others as deemed appropriate. Currently, it is quite difficult to figure out at the first read what was done and what are the core findings.

DISCUSSION

I suggest putting limitations of the study at the end of the discussion.

Please provide information on how your results will impact research and/or clinical practice.

Please discuss the generalizability of the results to the wider TKA population.

Author Response

Response to Reviewer 2 Comments

ABSTRACT

The link between background information and study aims is not clear.

Response: First of all, we really thank the reviewer for taking time and making effort to review our paper. Peripheral nerve block (PNB) for patients with total knee arthroplasty (TKA) is one of the recommended interventions in ERAS protocols. In order to clarify the relevance to the study aims, the status of the currently published studies is presented first. Change can be seen in line 18-21.

INTRODUCTION

Please start the first sentence in the following way and cite suggested papers:

“Osteoarthritis (OA) is the most common rheumatic disease of the musculoskeletal system, with the knee being the most commonly affected joint . The predominant clinical symptoms of knee OA are pain, joint stiffening, creptitus (https://pubmed.ncbi.nlm.nih.gov/12603937/), inflammation and muscle weaknes (https://pubmed.ncbi.nlm.nih.gov/28929165/ ). Established conservative treatement for osteoarthritis include exercise (https://www.ncbi.nlm.nih.gov/pmc/articles/PMC3635671/ ), knee bracing (https://pubmed.ncbi.nlm.nih.gov/29931372/ ), physical modalities  ( https://pubmed.ncbi.nlm.nih.gov/25162407/ ), pharmacology (https://www.ncbi.nlm.nih.gov/pmc/articles/PMC6315310/ ), but their long-term effectiveness is limited, thus total knee replacement is ultimately needed for the majority of patients with knee OA. “

Response: Thanks for the great suggestion from the reviewer. We have revised the first paragraph to reflect the suggested opinions. Change can be seen in line 38-44.

In the current form, it is quite difficult to figure out from the information flow in the introduction, why it is important to study this, who will benefit from it, and what is the added value of this paper to current knowledge.

Response: It is important as part of the ERAS program to find out how effective PNB is for both UTKA and BTKAs. Through this study it will be possible to apply appropriate postoperative management to patients who underwent TKA. Also, we will provide protocols that can be newly applied to the ERAS program. Change can be seen in line 70-79.

METHODS

The whole methods section should be rewritten according to relevant reporting guidelines, such as STROBE or CONSORT. Information on study design, setting, inclusion and exclusion criteria for study participants, the definition of outcomes, validity and reliability of instruments, data postprocessing etc is very superficial and should be provided in separate paragraphs to facilitate reading.

Response: Based on the suggestions given, we rearranged whole methods section. In addition, by separating the paragraphs according to each factor. To facilitate the validity and reliability of instruments, a description of the propensity score matching performed to control selection bias was added (Change can be seen in line 70-79). Because it was a retrospective study, The explanation of how the sample size was arrived at and how quantitative variables were handled in the analyses were insufficient.

RESULTS

Can authors better structure the results section? Given the amount of information, I would suggest providing subsections based on the outcomes or others as deemed appropriate. Currently, it is quite difficult to figure out at the first read what was done and what are the core findings.

Response: We redesigned results section structure for make it easier to figure out the results. In addition, a summary was added to the first sentence of each paragraph for easy understanding of the results.

DISCUSSION

I suggest putting limitations of the study at the end of the discussion.

Response: We really thank the reviewer for the good suggestion. We have adjusted the position of the constraint paragraph.

Please provide information on how your results will impact research and/or clinical practice.

Response: In consideration of your suggestions, we added the clinical impact of our results. it is thought that PNB can be actively applied clinically not only in UTKA but also in BTKA. Additionally, PNB was shown to have a better effect on pain control in UTKA than BTKAs. The cause of these results can be thought to be that in the case of BTKAs, the continuous dose and the single shot block were applied to each knee, respectively. Change can be seen in line 313-319

Please discuss the generalizability of the results to the wider TKA population.

Response: We discussed the generalizability of the results regard to the expansion of the scope of application of PNB. It is thought that it can be applied to revisional TKA. Change can be seen in line 321-323.

Round 2

Reviewer 1 Report

Thanks for responses and revision.

Reviewer 2 Report

I thank the authors for their response. The authors fully addressed my comments.